# Active droploids

Jens Grauer[1,4], Falko Schmidt[2,4], Jesús Pineda[2], Benjamin Midtvedt[2], Hartmut Löwen[1], Giovanni Volpe[2] & Benno Liebchen[3✉]

Active matter comprises self-driven units, such as bacteria and synthetic microswimmers, that can spontaneously form complex patterns and assemble into functional microdevices. These processes are possible thanks to the out-of-equilibrium nature of active-matter systems, fueled by a one-way free-energy flow from the environment into the system. Here, we take the next step in the evolution of active matter by realizing a two-way coupling between active particles and their environment, where active particles act back on the environment giving rise to the formation of superstructures. In experiments and simulations we observe that, under light-illumination, colloidal particles and their near-critical environment create mutually-coupled co-evolving structures. These structures unify in the form of active superstructures featuring a droplet shape and a colloidal engine inducing self-propulsion. We call them active droploids—a portmanteau of droplet and colloids. Our results provide a pathway to create active superstructures through environmental feedback.

[1] Institut für Theoretische Physik II: Weiche Materie, Heinrich-Heine-Universität Düsseldorf, D-40225 Düsseldorf, Germany. [2] Department of Physics, University of Gothenburg, SE-41296 Gothenburg, Sweden. [3] Institut für Physik der kondensierten Materie, Technische Universität Darmstadt, D-64289 Darmstadt, Germany. [4] These authors contributed equally: Jens Grauer, Falko Schmidt. ✉email: benno.liebchen@pkm.tu-darmstadt.de

Active matter consists of microscopic objects which are capable of self-propulsion, such as motile microorganisms, like *Escherichia coli* bacteria[1], and synthetic colloidal microswimmers, like autophoretic Janus colloids[2,3]. In contrast to equilibrium systems, active-matter systems evade the rules of equilibrium thermodynamics by continuously dissipating the energy obtained from their environment and converting part of it into a persistent motion at the level of individual constituents. Thus, the environment acts as a persistent free-energy source that emancipates active systems from the fundamental constraint of entropy maximization (or free-energy minimization) and induces a rich phenomenology, which is fundamentally beyond equilibrium physics. This includes, in particular, the spontaneous emergence of spatiotemporal patterns[3,4], e.g., through liquid–liquid phase separation[5–7], which illustrates the remarkable fact that driving systems far away from equilibrium often creates structures, not chaos.

Two key examples of self-driven agents are active colloids[8] and active droplets[9,10]. These are ideal model systems providing fundamental insights into the principles that govern the dynamical behavior of self-organized structures[11,12]. Active colloids catalyze reactions on part of their surface resulting in self-propulsion and collective self-organization as, e.g., living clusters[2,13–18], swarms,[19,20] and phase-separating macrostructures[21]. Active droplets exhibit complex dynamical behavior down to the level of individual droplets, where, e.g., chemical reactions drive them out of equilibrium and can influence the droplet formation process through growth suppression[22] or spontaneous droplet division[23]. Recent studies have also demonstrated internally propelled droplets, where a dense suspension of motile bacteria encapsulated in an emulsion droplet is able to transfer activity to the droplet, making it active[24,25].

In all these examples, and other typical active-matter systems, the environment serves as a continuous free-energy source which can also mediate effective interactions between active agents, such as hydrodynamic interactions synchronizing filaments[26] or interactions based on visual perception[27], acoustic signals[28], or chemical fields[29]—but it does not typically show intrinsic dynamics that adapts to the dynamics of the active agents. In contrast, biological systems often feature a two-way coupling with their environment, which is involved, e.g., in homeostasis, gene-expression regulation, and structure formation[30,31], calling for synthetic realizations to enable a controllable exploration of two-way feedback coupled systems.

Here, we experimentally realize and theoretically model a versatile feedback loop between light-activated colloids and their near-critical environment. This leads to structure formation both on the level of the colloids and their environment and results in the formation of a new type of self-propelling superstructures.

As these active superstructures combine droplets and colloids, we name them active droploids. To realize this feedback loop and the corresponding structure formation processes in the environment, we suspend a mixture of colloids in a near-critical solvent. When illuminated by laser light, these colloids absorb the light, locally heat up their surrounding environment and induce a local phase separation, which leads to the emergence of droplets around the colloids (see Fig. 1a). In turn, these droplets encapsulate the colloids, which under confinement interact non-reciprocally and form self-assembled colloidal engines, which finally drive the droploid superstructures, making them active (see Fig. 1a). Note that, crucially, both the colloids and the droplets continuously evolve in time, rather than adiabatically following the respective other component, which is fundamental to the droploids' structure formation and self-propulsion ability. Our findings offer a novel route to create superstructured soft active materials whose size and motility is controllable by laser light. Since it is based on the two-way interaction between

colloidal particles and a near-critical environment, the involved mechanism of colloidal self-encapsulation and activation might also serve as a useful framework to recreate and explore aspects of fundamental biological processes in a well-controllable synthetic colloidal minimal system. Examples might comprise, e.g., processes which are involved in the compartmentalization of the cytoplasm and the formation of membrane-free organelles, which share various features with liquid droplets and do not need a stabilizing lipid bilayer to maintain themselves.

## Results

**Experimental observations of active droploids**. We study a system of hydrophilic colloidal particles (radius $R = 0.49\,\mu m$) quasi-two-dimensionally confined between two glass slides separated by a distance smaller than two particle diameters. These particles are immersed in a near-critical water–2,6-lutidine mixture, which has a critical lutidine composition $c_c^L = 28.4\%$ and a lower critical temperature $T_c = 34.1\,°C$[32]. In this work, we use a slightly off-critical composition (29.4–32.4%) to ensure the formation of water-rich droplets around the hydrophilic particles when the mixture's temperature locally exceeds $T_c$. We fix the temperature of the sample at $T_0 = 32.5\,°C < T_c$ using a water heatbath and a feedback temperature controller (see experimental setup in Supplementary Fig. 2).

We use two species of hydrophilic particles: light-absorbing and non-absorbing particles. In the absence of an external light source, both species perform passive Brownian motion with a diffusion coefficient of $D = 0.012 \pm 0.002\,\mu m^2\,s^{-1}$ and are homogeneously distributed in the sample chamber (Fig. 1b). The non-absorbing particles are less hydrophilic than the absorbing ones.

Under illumination with a defocused laser beam ($\Lambda = 1070$ nm, $I = 142\,\mu W\,\mu m^{-2}$, beam waist $w = 100\,\mu m$), light-absorbing particles raise the temperature of the surrounding liquid slightly above $T_c$, thus altering their local environment by inducing a local demixing of the liquid. This leads to the creation of local gradients of the mixture's temperature and composition, which phoretically attract other particles present nearby[33–36]. In particular, this generates a non-reciprocal effective attraction of the non-absorbing particles by the absorbing ones leading to ballistically moving active molecules (Fig. 1c), Janus-dimers being the simplest example[36].

At comparatively large light intensities, where the system of active colloids induces local temperatures exceeding the critical temperature (i.e., $T \gg T_c$, Fig. 1d), we observe a stronger feedback between the particles and the environment: the absorbing colloids induce phase separation in their vicinity, which leads to the confinement of the active colloidal molecules within water-rich droplets immersed in a lutidine-rich background. Remarkably, we observe that these droplets can adopt the mobility of the active molecules which they comprise. This occurs because the colloidal molecules contained within a droplet constantly alter their local environment causing the droplet to follow the molecules' motion. Moreover, we find that the molecules' direction inside the droplet is reversed leading now with absorbing particles in front due to the local changes in composition[37]. In this state, we observe the emergence of active droploids. Once formed, these active droploids move, collide, and merge with each other and consequently grow over time (Fig. 1e), until they eventually all coalesce into a large active droploid (Fig. 1f).

**Model and simulations**. Let us now build a minimalist theoretical model to identify the key ingredients and mechanisms determining the experimental observations described in the previous section. This model describes the combined dynamics and

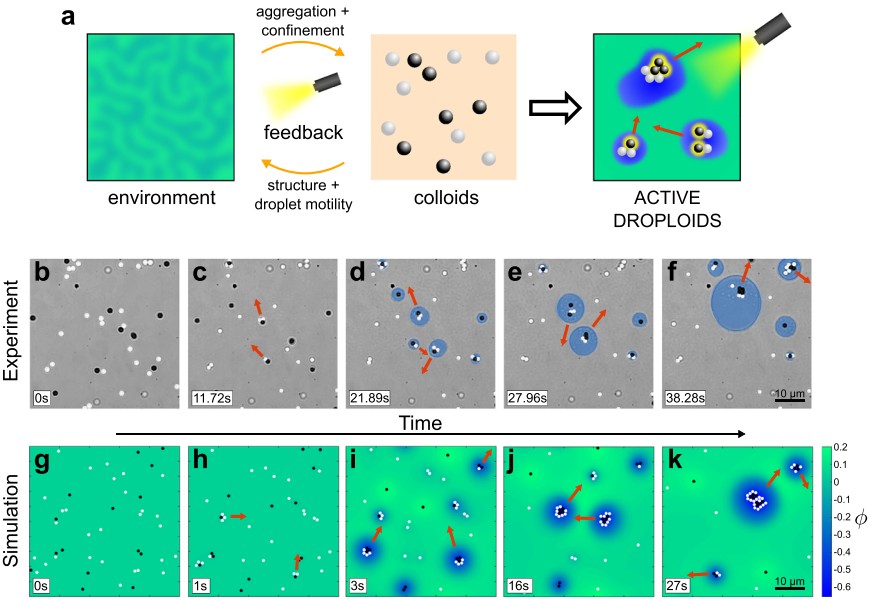

**Fig. 1 Active droploid formation and growth. a** Schematic of the light-induced two-way coupling (feedback loop) between the colloids and their environment which results in active droploids. **b–f** Experiment and **g–k** simulation of the formation and growth of active droploids. **b, g** Single particles of two species, light-absorbing (black) and non-absorbing (white), are immersed in a near-critical mixture and, at low temperature, behave as passive particles in a standard liquid. **c, h** Upon illumination, absorbing particles heat up the surrounding liquid, providing phoretic forces that bring and hold particles together to form small colloidal molecules that move in the direction of the red arrows. **d, i** Eventually, local phase separation leads to water-rich droplets (blue shading) surrounding absorbing particles and colloidal molecules. **e, j** Over time, the active droploids move together with their active molecules (direction indicated by the red arrows), grow in size, and **f, k** eventually coalesce together to form even larger active droploids. The light irradiation is $I = 150\,\mu W\,\mu m^{-2}$, composition $\phi_0 = 0.05$ and initial temperature $T_0 = 32.5\,°C$ (and $\lambda = 0.025$ in simulations, see 'Methods' for other parameter values). Videos of experiment (Supplementary Movie 1) and simulation (Supplementary Movie 2) are provided in SI.

feedback between the near-critical mixture and the colloidal particles.

The state of the mixture is defined by the order parameter field $\phi(\mathbf{r}, t)$, which represents the relative concentration difference between the two phases of the mixture: phase $A$ (2,6-lutidine) and phase $B$ (water). Thus, we have $\phi = 0$ in regions where $A$ and $B$ are homogeneously mixed, and $\phi = \pm 1$ in pure $A$ and $B$ regions, respectively.

The dynamics of the colloids is modeled as overdamped Brownian particles at positions $\mathbf{r}_i^s(t)$ following Langevin dynamics ($i = 1, \ldots, N$, where $N$ is the number of particles and $s \in \{a, na\}$ for absorbing and non-absorbing particles, respectively)

$$\gamma \partial_t \mathbf{r}_i^s(t) = \beta_s \nabla \phi + \alpha_s \nabla(\nabla \phi)^2 - \nabla_{\mathbf{r}_i} V + \sqrt{2D}\gamma \boldsymbol{\eta}_i^s , \quad (1)$$

where $D$ is the translational diffusion coefficient of the particles, $\gamma$ is the Stokes drag coefficient (assumed to be the same for both equally sized species), $\boldsymbol{\eta}_i^s(t)$ represents Gaussian white noise with zero mean and unit variance, and $V$ accounts for steric repulsions between the colloids, represented by Weeks-Chandler-Anderson (WCA) interactions[38]. The coupling to the composition field $\phi(\mathbf{r}, t)$ (environment) is described by the first two terms on the RHS of Eq. (1). The first term describes the net effect of wetting, i.e., the fact that hydrophilic particles ($\beta_s < 0$) are attracted by water-rich droplets, whereas hydrophobic particles ($\beta_s > 0$) tend to remain outside of these regions (see Supplementary Fig. 3). The second term, which is proportional to $\nabla(\nabla \phi)^2$, induces motion towards interfaces (where $(\nabla \phi)^2$ is large) essentially for the non-absorbing particles. This term describes the tendency of the weakly hydrophilic particles to move towards the water–lutidine interface in order to reduce the interfacial area of the water–lutidine interface and hence the total interfacial free energy of the system.

To model the phase separation dynamics induced by the light-absorbing particles, we use the Cahn-Hilliard equation[39] taking

into account an inhomogeneous temperature distribution $T(\mathbf{r}, t)$ as induced by the light-absorbing particles

$$\partial_t \phi(\mathbf{r}, t) = M\nabla^2 \left( a(T - T_c)\phi + b\phi^3 - \kappa \nabla^2 \phi + A_s \sum_{s \in \{a, na\}} \delta(\mathbf{r} - \mathbf{r}_i) \right) , \quad (2)$$

where $M$ is the inter-diffusion constant of the mixture, and $T_c$ is the critical temperature, with constants $a < 0$ and $b, \kappa > 0$ such that the fluid demixes at locations where $T > T_c$. To describe the net effect of the accumulation of water at the hydrophilic surfaces of the colloids, we include a (point-like) source term for the solvent-composition at the position of each particle. The coefficients $A_a$ and $A_{na}$ are chosen such that they account for the strong and weak hydrophilicity of the absorbing and non-absorbing particles, respectively. As a result, the water concentration slightly increases at the location of each particle. Note that this increase alone cannot initiate phase separation, but it biases the location where water-rich droplets occur once phase separation takes place.

The two-way coupling between the nonequilibrium system of particles and its environment is controlled by the mixture concentration and the energy supply. The former is given by the order parameter field $\phi(\mathbf{r}, t)$, described above. The latter depends on the density of absorbing particles $\rho_a$ and the light intensity $I(\mathbf{r})$, and involves a suitable source term for the absorbed power per unit volume $\frac{\alpha'}{\rho c_p} I(\mathbf{r}) \equiv k_0 \delta(\mathbf{r} - \mathbf{r}_i)$, where $\alpha'$ is the optical absorption coefficient, $\rho$ is the density of the mixture, $c_p$ is the specific heat at constant pressure, and $k_0$ is the strength of the light source at the particle position $\mathbf{r}_i$[40]. The inhomogeneous temperature field is then to be calculated from the heat equation as

$$\partial_t T(\mathbf{r}, t) = D_T \Delta T + k_0 \sum_{\text{absorb.}} \delta(\mathbf{r} - \mathbf{r}_i) - k_d(T - T_0) \quad (3)$$

with diffusion constant $D_T$. Here, the decay rate $k_d$ describes the

coupling of the sample to an external water heatbath stabilizing the temperature (see SI for a detailed description of the experimental setup and methods for the equations of motion). Overall, this permits us to introduce a concise measure of the energy input under the approximation that the adsorbed energy scales linearly with the density of the absorbing particles and the irradiated light intensity as $\lambda = \frac{\rho_a k_0}{T_0 k_d}$.

Using the model we have described, we can investigate the complex dynamics involved in the formation of droplets and molecules including the involved feedback loop between the colloids and their near-critical environment. In particular, we are able to replicate in simulations all experimentally observed states, i.e., from passive disperse particles to ballistically moving active droploids (Fig. 1g–k).

**Phase diagram**. By continuously altering their local environment, absorbing particles create feedback loops that, depending on the criticality of the environment and the energy input to the system, induce assembly and disassembly of colloidal molecules and determine the dynamics of the colloid–droplet superstructure. To gain a systematic overview of the possible states achievable by this system, we now determine the full nonequilibrium state-diagram as a function of the composition order parameter $\phi_0$ (see also phase diagram in Supplementary Fig. 1) and of the measure of the energy input into the system $\lambda$ (see previous section). The resulting phase diagram after 30 s of light illumination is shown in Fig. 2 for $\phi_0 > 0$ (i.e., at supercritical 2,6-lutidine concentrations, $c^L > c_c^L$, leading to water-rich droplets in a lutidine-rich background). Similar and symmetric results can be observed for $\phi_0 < 0$ (i.e., $c^L < c_c^L$, where lutidine-rich droplets emerge in a water-rich background). Note that the different structures can dynamically move, coalesce, and grow over time (see also Fig. 3), but determining the different phases at much later times qualitatively produces the same phase diagram (see Supplementary Fig. 5).

As can be seen in Fig. 2a, we identify four distinct states differing in their level of activity and in the presence of droplets. At low energy input and at concentrations far away from the critical composition, we observe a disordered phase (purple region in Fig. 2a), which is characterized by randomly dispersed Brownian particles essentially behaving as passive particles at thermodynamic equilibrium (Fig. 2b, purple frame). Increasing the energy input, we observe an active molecules' phase (yellow region in Fig. 2a), where active and passive colloids come together to form active colloidal molecules (Fig. 2b, yellow frame).

The remaining two phases of the phase diagram are located at even higher energy inputs. In these cases, the temperature around absorbing particles and active colloidal molecules significantly exceeds the critical temperature ($T > T_c$), which induces a local phase separation of the mixture and results in the formation of water-rich droplets surrounding the absorbing particles and colloidal molecules. Subsequently, nearby colloids are absorbed into the droplet due to their own hydrophilicity, so that the colloidal molecules inside the droplets grow in size over time. This procedure permits a good observation of the influence of colloids on their environment, which deform the interface when entering the droplet or while moving alongside it (see Fig. 4a, b), which has also been observed for vesicles[41]. At a moderate energy input, we observe the active droploids' phase in our phase diagram (green region in Fig. 2a): the active colloidal molecules contained within a droplet manage to propel the droplet (Fig. 2b, green frame); thus, the droplets become active by themselves. Thus, active colloidal molecules contained inside a droplet act as internal motors that propel the droplet. Over time, these active droploids can collect other molecules and droplets, thereby growing in size and possibly altering their speed and direction of movement (see Supplementary Movies 1 and 2). The speed and growth process of the droplets can be controlled by light intensity as discussed in the following section.

At even higher energy inputs (achievable either by increasing the light intensity or by a higher density of absorbing particles), the

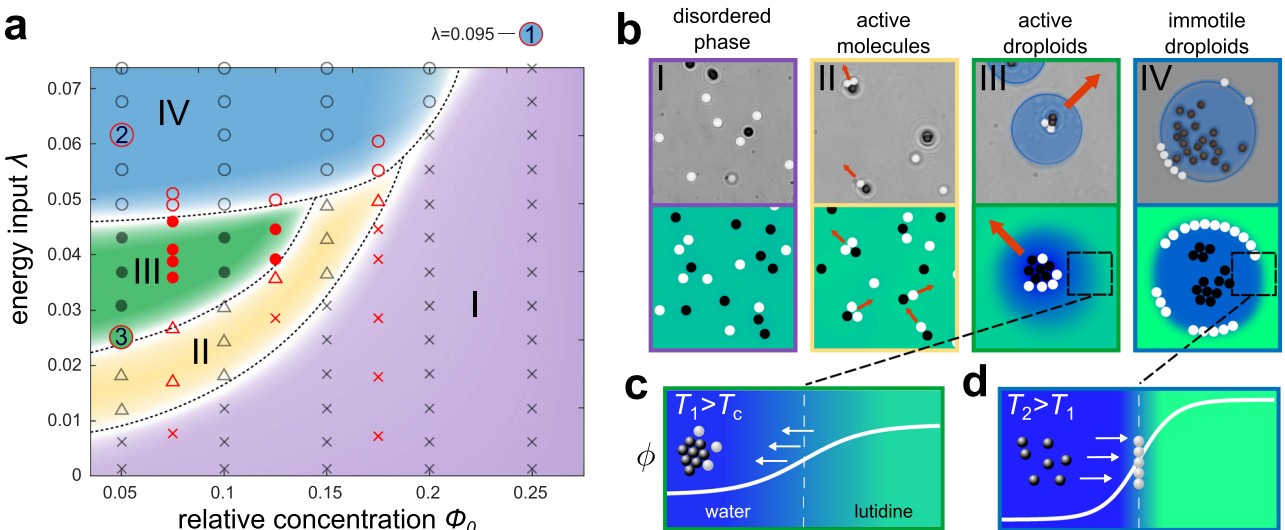

**Fig. 2 Nonequilibrium phase diagram. a** Phase diagram as a function of the net energy input $\lambda$ and the averaged relative concentration difference from the critical point $\phi_0$. The evaluated state points from the experiment (red) fitted with $\lambda = C l \rho_a$ and $C = 1 \times 10^{-4}\,\mu m^4\,\mu W^{-1}$, and the simulations (black) are indicated by crosses (purple region - disordered phase), triangles (yellow region - active molecules), filled circles (green region - active droploids), and empty circles (blue region - droplets with particles at the interface). Dashed lines indicate approximate boundaries between phases and serve as a guide to the eye. The quantitative criteria for the phases and the corresponding colors are given in the SI. The red-bordered numbers mark reference points that relate to different scenarios discussed in the main text. **b** Typical snapshots from experiments (top) and simulations (bottom) of the phases I–IV as indicated in the state diagram. Magnified concentration profiles of the composition in phases III (**c**) and IV (**d**) show that the gradient at the interface steepens as temperature locally increases from $T_1 > T_c$ (active droploids, phase III) to $T_2 > T_1$ (immotile droploids, phase IV). Simulation parameters can be found in 'Methods'.

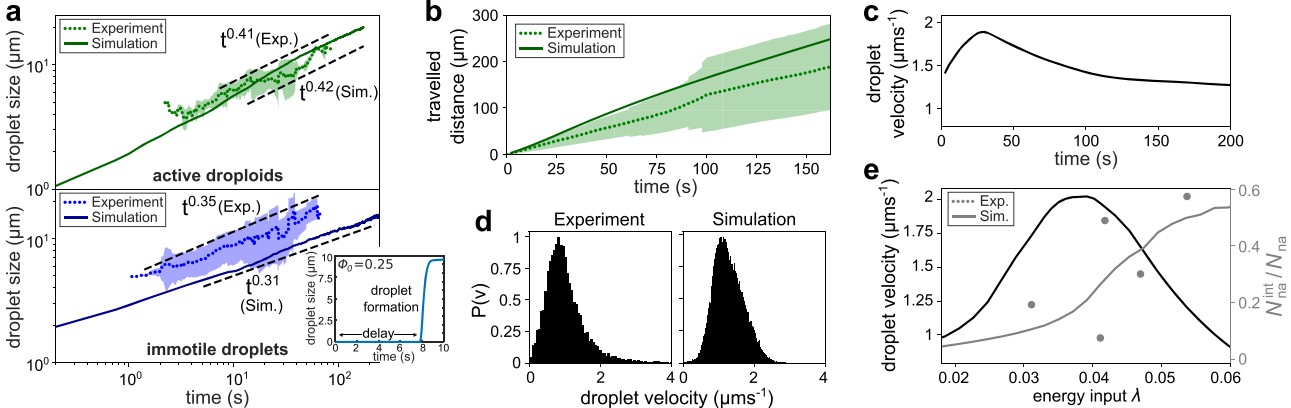

**Fig. 3 Droplet velocity and growth over time. a** Average size of active droploids (green) and immotile droplets (blue) over time calculated from experiments (dotted) and simulations (solid) for $\phi_0 = 0.05$. The shaded area represents the standard deviation. The inset shows the delay in the formation of an immotile droplet at early times for an off-critical composition of $\phi_0 = 0.25$. **b** Mean (and shaded standard deviation) of the total traveled distance of an active droploid over time measured from experiments (dotted) and simulations (solid). **c** Simulated active droploid velocity over time. **d** Velocity distribution of active droploids in experiments (left) and simulations (right). **e** Simulated mean velocity of active droploids after 30 s of light illumination (black curve) and fraction of non-absorbing particles located at the interface of the droplets $N_{na}^{int}/N_{na}$ (gray curve simulations, gray dots experimental data fitted with $\lambda = Cl\rho_a$ and $C = 3 \times 10^{-4}\ \mu m^4\ \mu W^{-1}$) as a function of the energy input $\lambda$ for $\phi_0 = 0.05$. Note that the fitting factor $C$ is different here than in Fig. 2 because the present measurements are based on a fixed concentration of $\phi_0 = 0.05$, whereas those in Fig. 2 have been taken at various concentrations up to $\phi_0 = 0.2$. The full list of simulation parameters is provided in the 'Methods' section.

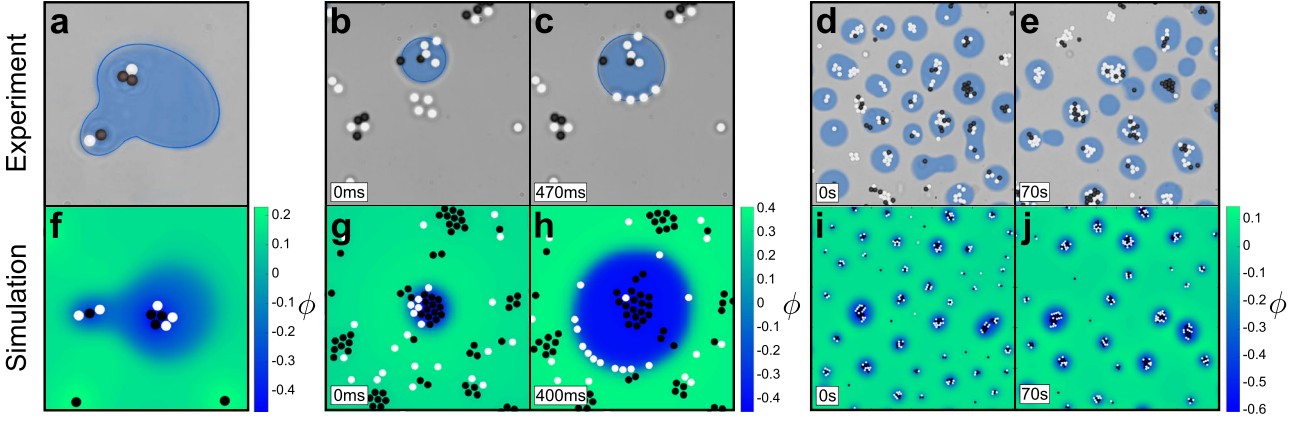

**Fig. 4 Examples of droplet behavior. a–e** Experimental snapshots (highlighting the segmentation of the droplets), and **f–j** simulated snapshots (background displays the relative concentration $\phi$) of various behaviors: **a**, **f** Clusters of colloids deforming the boundary of the droplet at $\lambda = 0.03$. **b**, **c** and **g**, **h** Accumulation of absorbing particles in an off-critical supersaturated background phase ($\phi_0 = 0.2$ in experiments, $\phi_0 = 0.25$ in simulations) leading to an explosive formation of droplets within a very short time at $\lambda = 0.095$ (Supplementary Movies 3 and 4). **d**, **e** and **i**, **j** Formation of size-stabilized droplets around absorbing particles with periodic light illumination (on and off for 10 s each, Supplementary Movies 5 and 6) at $\lambda = 0.025$. Other simulation parameters can be found under 'Methods'.

induced temperatures greatly exceed the critical temperature $(T \gg T_c)$ and result into a large phoretic gradient close to the interface of the droplet (Fig. 2d) compared to active droplets where the gradient is more moderate (Fig. 2c). Because the dynamics of the particles is mainly controlled by the interaction of their surface with the local composition of the mixture, non-absorbing particles, which are less hydrophilic than their absorbing counterpart, move towards the droplet's interface to reduce the total interfacial area of the system (Fig. 2b, blue frame). Consequently, the existing colloidal molecules break up with absorbing particles remaining at the center of the droplet and non-absorbing particles decorating its interface. The ensuing loss of motility characterizes this phase as the immotile droploids' phase (blue region in Fig. 2a).

Overall, the dynamics shown by this phase diagram shows that absorbing particles continuously alter their local environment, and, in turn, their behavior is affected by the environment. This feedback induces both assembly and disassembly of active

colloidal molecules and, therefore, determines the ensuing activity of the whole system of colloids and droplets.

**Characterization and control of droplet dynamics and growth**. By tuning the external energy input and the mixture's criticality, we can control the degree of interaction between active colloids and their local environment, which determines the overall state of the system (Fig. 2) as well as its evolution over time.

We start by characterizing the overall growth of our system (Fig. 3a). At early times, the system is characterized by nucleation and formation of droplets initiated by light-absorbing particles. In this initial process, the droplets slowly grow over time as additional colloidal molecules diffuse from the bulk phase and contribute to the local heating that creates the droplet. The dominant growth process is therefore diffusion-limited and a droplet diameter $L \sim t^{\frac{1}{2}}$ is expected[7].

Using a composition of $\phi_0 = 0.25$ (marked with a red-bordered 1 in Fig. 2) which is far away from the critical composition, we observe that droplets form only after a significant initial delay. For off-critical compositions, the bulk phase is already supersaturated and a considerable (free-energy) barrier emerges, separating the mixed phase from complete phase separation (see Supplementary Fig. 1b). Consequently, a large accumulation of absorbing particles is required to provide sufficient local energy to overcome this barrier and cause nucleation, resulting in a ballistic formation of a droplet. This means that droplets do not form immediately, but with a slight time delay, and then grow rapidly in size. The size of such a droplet is shown in the inset of Fig. 3a, where it takes 8 s after the critical temperature has already been exceeded to allow the formation of a droplet. We observe such an explosive droplet formation both in experiments (Fig. 4b, c and Supplementary Movie 3) and in simulations (Fig. 4g, h and Supplementary Movie 4). Droplet nucleation can be further delayed if other nearby absorbing particles also form clusters or even droplets, which then compete with each other as the concentration of the droplet's phase (here water) is locally decreased inside the bulk mixture.

For concentrations around $\phi_0 = 0.05$ (marked with a red-bordered 2 in Fig. 2a), after the droplets have formed, we observe a transition from nucleation and growth to coarsening. Such a late-time coarsening regime is expected, determined by Brownian coalescence of droplets and diffusion-limited coarsening. For Brownian coalescence, small droplets collide with each other and fuse to form larger droplets, reducing the overall interface. For diffusion-limited coarsening, the dominant growth process is given by the transport of droplet-forming molecules from small droplets into large droplets growing by diffusion from the bulk phase. For both coalescence and coarsening, $L \sim t^{\frac{1}{3}}$ is expected[7,42–47], which we can also find in our experiments ($L \sim t^{0.35}$) and simulations ($L \sim t^{0.31}$) by measuring the average size of non-moving droplets over time (blue curves in Fig. 3a), and that are passive because, either the number of non-absorbing particles is small, or these particles are concentrated at the droplet's interface.

The picture is different if the droplets feature ballistic movement driven by the presence of active colloidal molecules within the droplets themselves. In fact, the self-propulsion of active droploids accelerates the growth process described above. We observe that the size of active droploids grows as $L \sim t^{0.42}$ (marked with a red-bordered 3 in Fig. 2a and shown in Fig. 3a), which is significantly faster than the observed $L \sim t^{0.31}$ growth law for passive droplets. The accelerated growth is a direct consequence of the ballistic motion of the active droploids which allows them to recruit additional colloids faster than the passive droplets. This can be easily seen by measuring the total distance traveled in time by an active droploid (shown in Fig. 3b for experiment and simulation), revealing that the ballistically moving active droploids are able to cover a comparatively large area allowing them to efficiently collide and fuse. The enhanced growth process is depicted by the average size of the droplet domain in the experiments and the simulations in Fig. 3a (green curves), which is close to the expected $L \sim t^{\frac{1}{2}}$ growth law for ballistic aggregation[48,49]. As shown in Fig. 3c, the active droploids slow down at later times. Ultimately, they would reach a state where light-absorbing and non-absorbing particles form a major cluster of almost randomly arranged particles within the droplet shell, resulting, for statistical reasons, in a reduced self-propulsion[36]. The growth of the colloidal clusters within the droplet shells also enhances the temperature locally, which increases the degree of demixing and provides a further reason for the slow-down of the droploids.

Since the self-propulsion of a droploid is determined by the number and composition of the contained colloidal molecules, we have explored the velocity distribution of the active droploids in our experiments and simulations (Fig. 3d), revealing, in particular, a small positive skewness of the distribution in both cases, indicating a tail towards large velocities.

Furthermore, we can additionally control the droplet velocity by light intensity. In the parameter regime in which active droploids can be found (Fig. 2, phase III, $0.023 < \lambda < 0.048$), an increase in light intensity leads to an increase in droplet speed (Fig. 3e). For increasing intensities, however, the velocity reduces as the resulting temperature in the sample increases and non-absorbing particles accumulate at the water–2,6-lutidine interface (transition from green to blue in the state diagram in Fig. 2a). Consequently, molecules inside the droplet slowly dissolve. We can characterize this transition counting the number of non-absorbing particles located at the interface $N_{\mathrm{na}}^{\mathrm{int}}$ as a fraction of the total number of non-absorbing particles $N_{\mathrm{na}}$ (see gray line in Fig. 3e). Between $\lambda = 0.037$ and 0.045, the fraction of non-absorbing particles at the interface significantly increases from $N_{\mathrm{na}}^{\mathrm{int}}/N_{\mathrm{na}} = 0.15$ to 0.35, whereas the growth in the droplet's velocity has slowly decreased with reaching its maximum velocity of $v = 2 \, \mu m \, s^{-1}$ at $\lambda = 0.038$. For larger values of $\lambda$, the number of non-absorbing particles at the interface is sufficiently large to rapidly decrease the droplet's velocity and finally reach a value similar to that of immotile droplets ($v < 0.8 \, \mu m \, s^{-1}$).

While droplet speed and the growth of droplets can be accelerated by increasing the laser intensity, the growth process can also be arrested by periodic light illumination. Employing periodic illumination, where the light is alternately switched on and off for a duration of 10 s (0.1 Hz), we show that the further growth of droplets can be slowed down and even arrested (Fig. 4d, e and Supplementary Movie 5), which is in good agreement with simulations (Fig. 4i, j and Supplementary Movie 6). During times of no illumination, temperatures quickly drop below $T_c$, droplets dissolve and colloidal molecules disassembly as their constituent particles diffuse apart. Upon illumination, this process is reversed and colloid–droplet superstructures reappear. This shows that by adjusting light illumination, we achieve temporal and spatial control over the system of colloids and droplets.

## Discussion

Our results show that a two-way coupling between the motion of colloidal particles and the dynamics of their environment creates a route towards a novel class of active superstructures. These structures hinge on mutually coupled structure formation processes of the colloids, which form an engine, and the surrounding solvent, which phase separates in regions of high colloidal density and encapsulates the engine within a droplet shell. Our results create a bridge between the physics of active colloids and droplets and provide fundamental insights into the role of feedback for the emergence of ordered active superstructures, which opens up new possibilities for active-matter research to investigate two-way feedback loops in other systems and to create light-activated biomimetic materials.

## Methods

**Experimental setup**. We consider a suspension of colloidal particles in a critical mixture of water and 2,6-lutidine at the critical lutidine mass fraction $c_c^L = 0.286$ with a critical temperature at $T_c = 34.1 \, °C^{[50]}$ (see Supplementary Fig. 1a). The light-absorbing particles consist of silica microspheres with light-absorbing iron-oxide inclusions (microParticles GmbH), while the non-absorbing particles consists of equally sized plain silica microspheres (microParticles GmbH). Both particle species possess the same radius ($R = 0.49 \pm 0.03 \, \mu m$) and have similar density ($\rho \approx 2 \, g \, cm^{-3}$). The suspension is confined in a sample chamber quasi-two-dimensionally between a microscope slide and a cover slip, where the particles are sedimenting to due to gravity. We use spacer particles (silica microspheres, microParticles GmbH) with a radius $R = 0.85 \pm 0.02 \, \mu m$ for constant separation but with a concentration $c \ll 5\%$, in order to not interfere with the observed

phenomena. We have treated our glass surface prior with NaOH solution ($c = 1$ mol) creating a smooth hydrophilic layer on top. Surprisingly, we found that a particle solution prepared at $c_c^L$ in such a sample chamber behaved off-critical (i.e., nucleation of droplets). By adding about 2% more water to the mixture critical behavior returned (i.e., spinodal demixing). We expect that the hydrophilic surfaces of the sample chamber reduced the bulk concentration of water for which we have compensated.

A schematic of the setup is shown in Supplementary Fig. 2. The particle motion is captured by digital video microscopy at 12 fps. Using a two-stage feedback temperature controller[50,51], the sample's temperature is kept near-critical at $T_0 = 32.5$ °C, where water and 2,6-lutidine are homogeneously mixed. Under these conditions, the microspheres of both species are passive immotile Brownian particles performing standard diffusion (Fig. 1b, g). The sample is illuminated from above using a defocused laser of wavelength $\Lambda = 1070$ nm at varying intensities. The increase of temperature surrounding the light-absorbing particles is rather small ($\Delta T \approx 2$ °C) such that they still behave as non-active Brownian particles.

The segmentation of the droploids is made using a deep neural network implemented and trained using DeepTrack 2.0[52] (see details in SI and also Supplementary Movie 7).

**Details on the simulation model**. To model our experimental findings, we consider an ensemble of $N$ overdamped spheroidal colloids at position $\mathbf{r}_i$ immersed in a near-critical water–lutidine mixture, described by the Cahn-Hilliard equation, which can be derived from the total free-energy functional

$$\mathcal{F}[\phi] = \int d\mathbf{r}\left(\frac{a}{2}(T - T_c)\phi^2 + \frac{b}{4}\phi^4 + \frac{\kappa}{2}(\nabla\phi)^2 + \sum_{i=1}^{N}\phi V_{co}^s\right) \quad (4)$$

where $T_c$ is the critical temperature of the composition, with constant $a < 0$ and $b$, $\kappa > 0$ such that the fluid demixes, where $T > T_c$. Here we describe the coupling of the hydrophilic particles to the concentration of the mixture with an external potential which we approximate with $V_{co}^s(|\mathbf{r} - \mathbf{r}_i|) \approx A_s \delta(\mathbf{r} - \mathbf{r}_i)$, where $s \in \{a, na\}$ for absorbing and non-absorbing particles, respectively. The evolution of the conserved order parameter $\phi$ (composition of the two components) is then given by the Cahn-Hilliard equation

$$\partial_t \phi = M\Delta\frac{\delta\mathcal{F}[\phi]}{\delta\phi} \quad (5)$$

$$\partial_t \phi = M\nabla^2\left(a(T - T_c)\phi + b\phi^3 - \kappa\nabla^2\phi + A_a\sum_{absorb.}\delta(\mathbf{r} - \mathbf{r}_i) + A_{na}\sum_{non-abs.}\delta(\mathbf{r} - \mathbf{r}_i)\right) \quad (6)$$

where $M$ is the inter-diffusion constant of the mixture. We describe the impact of the hydrophilicity of the light-absorbing and non-absorbing particles on the dynamics of the fluid with an additional term including a $\delta$-function at the particle positions, whose strength is given by $A_a$ and $A_{na}$, respectively. The inhomogeneous temperature field produced by the light-absorbing particles with rate $k_0$ is to be calculated from the heat equation

$$\partial_t T(\mathbf{r}, t) = D_T\Delta T + k_0\sum_{absorb.}\delta(\mathbf{r} - \mathbf{r}_i) - k_d(T - T_0) \quad (7)$$

with decay rate $k_d$ and diffusion constant $D_T$. We can then phenomenologically describe the motion of the light-absorbing and non-absorbing particles $\mathbf{r}_i^s(t)$ ($i = 1, \ldots, N$, $s \in \{a, na\}$)

$$\gamma\partial_t\mathbf{r}_i^s(t) = \beta_s\nabla\phi + \alpha_s\nabla(\nabla\phi)^2 - \nabla_{\mathbf{r}_i}V + \sqrt{2D}\gamma\boldsymbol{\eta}_i^s \quad (8)$$

where $D$ is the translational diffusion coefficient of the particles, $\gamma$ is the Stokes drag coefficient (assumed to be the same for both species) and $\boldsymbol{\eta}_i^s(t)$ represents unit-variance Gaussian white noise with zero mean. Here, the first term describes the attraction into water-rich regions, caused by the particles' hydrophilicity, where the second term describes the tendency of the particles to attach themselves to the interface of the two components. In addition, $V$ accounts for excluded volume interactions among the particles which all have the same radius $R$ and which we model using the Weeks-Chandler-Anderson potential $V = \frac{1}{2}\sum_{i,j\neq i}V_{ij}$ where the sums run over all particles and where $V_{ij} = 4\epsilon\left[\left(\frac{\sigma}{r_{ij}}\right)^{12} - \left(\frac{\sigma}{r_{ij}}\right)^6\right] + \epsilon$ if $r_{ij} \leq 2^{1/6}\sigma$ and zero else. Here $\epsilon$ determines the strength of the potential, $r_{ij}$ denotes the distance between particles $i$ and $j$, $r_c = 2^{1/6}\sigma$ indicates a cutoff radius beyond which the potential energy is zero and $\sigma = 2R$ is the particle diameter.

**Simulation parameters**. In the simulation model we measure the distance in units of 1 μm and the time in units of 1 s and match the parameters such as diffusion constants, particle radius and typical velocity of the particles with the experiment. In all our simulations we use for the Cahn-Hilliard equation $M = 10^2$ μm² s⁻¹, $a = -2.5$ K⁻¹, $b = 50$, $\kappa = -5$ μm², $A_a = 2.5$ μm², $A_{na} = 1.5$ μm², for the dynamics of the heat equation $k_0 = 60 \times 10^3$ Kμm² s⁻¹, $k_d = 0.5 \times 10^3$ s⁻¹, $D_T = 10^4$ μm² s⁻¹, $T_0 = 32.5$ °C, and for the Langevin equation of the particles $\beta_a/\gamma = -0.5 \times 10^3$ μm² s⁻¹, $\beta_{na}/\gamma = -0.2 \times 10^3$ μm² s⁻¹, $\alpha_a/\gamma = 0$, $\alpha_{na}/\gamma = 1.5 \times 10^3$ μm⁴ s⁻¹, $D = 0.1$ μm² s⁻¹, and $\epsilon/\gamma = 100$ μm² s⁻¹.

Additionally we used in Fig. 1 $L_{box} = 50$ μm, $N_a = 15$, $N_{na} = 25$, $\phi_0 = 0.05$, and in Fig. 3 $L_{box} = 200$ μm, $N_{na} = 160$, $\phi_0 = 0.05$, and $k_0 = 80 \times 10^3$ Kμm² s⁻¹,

$N_a = 200$ (active droploids) and $k_0 = 50 \times 10^3$ Kμm² s⁻¹, $N_a = 800$ (immotile droplets) and in Fig. 4i, j we have $L_{box} = 100$ μm, $N_a = N_{na} = 160$, $\phi_0 = 0.05$ and $k_0 = 25 \times 10^3$ Kμm² s⁻¹.

## Data availability

All data are available from the corresponding author upon reasonable request.

## Code availability

The codes that support the findings of this study are available from the corresponding authors upon reasonable request.

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

## Acknowledgements
F.S., J.P., B.M., and G.V. acknowledge partial support by the ERC Starting Grant ComplexSwimmers (grant number 677511) and by Vetenskapsrådet (grant number 2016-03523). B.L. acknowledges support by the Deutsche Forschungsgemeinschaft (DFG, German Research Foundation) - Project number 233630050 (TRR-146).

## Author contributions
All authors have planned the project, have discussed the results, and have written or edited the manuscript. The experiments have been planned by F.S. and G.V. and the model, and the simulations by J.G., H.L., and B.L. The experiments and the simulations have been performed by F.S. and J.G., respectively. J. P. and B. M. have quantitatively tracked the colloids.

## Funding

## Competing interests
The authors declare no competing interests.
