## [Peer Review File. · Nature Communications]

Active DroploidsREVIEWER COMMENTS

Reviewer #1 (Remarks to the Author):

In the manuscript "Active droplids", the authors report on the realization of a two-way coupling between active particles and their environment, where active particles act back on the environment giving rise to the formation of superstructures that they call "droplids". Droplids are realized by light-activated colloids placed in a near-critical environment. When illuminated by laser light, these colloids absorb the light, locally heat up the surrounding environment, and induce a local phase separation leading to the emergence of droplets. The self-propulsion mechanism of symmetry-broken assemblies has been demonstrated earlier by the authors (J. Chem. Phys., 2019, 150, 094905) and others (Chem. Comm. 2018, 54, 11933). The authors employ these active assemblies to act as an engine in the droplet-colloids superstructures like the recently demonstrated active droplets (Soft Matter, 2020, 16, 1359 and Nat. Phys., 2021, 17, 260).

The authors experimentally realize droplids and theoretically model the observed phenomenology. The diagram of the dynamic states has been successfully captured by the simulations, as well as the aggregation dynamics of the droplets. The paper is clearly written, and conclusions seem to be well supported by the presented data.

At the end of the paper (conclusions section), the authors got a bit bombastic with the claim of "a novel class of active matter" associated with the droplids. Active droplets - combining droplets and active particles - have been around both in simulations and experiments for quite some time.

A few points that need further clarification:

In relation to the phoretic motion of non-absorbing particles discussed on Page 10 and Fig. 2c and 2d. Both absorbing and non-absorbing particles are hydrophilic. It is not clear why hydrophilic non-absorbing particles should go to the interface at elevated temperatures. Is it the difference in the degree of hydrophilicity between those two species of particles that promotes their motion towards the interface?

Also, what happens with droplids when you let the system relax below critical temperature?

Reviewer #2 (Remarks to the Author):

The authors report on the light-triggered assembly of colloidal clusters in near-critical environments. The manuscript focuses on describing the phenomenology of the system when the composition of the environment (its distance from the critical mixture) and the heat input are varied.

I have several reservations about the manuscript which in the present form lack the completeness expected for publication in Nature Communications. While the system displays rich behaviours, its characterization would deserve to be more thorough. Now, it appears somewhat like an addendum to a previous publication by the authors (Ref 31. <https://aip.scitation.org/doi/10.1063/1.5079861>).

- The very claim that no environment feedback was ever at play in active matter system (bottom of page 2) is inaccurate. What about the role of boundaries in active system (<https://www.nature.com/articles/nphys3876> for a recent example)? What about environment-mediated interactions between active particles (<https://doi.org/10.1103/PhysRevLett.123.208101> for a recent example)? The situation in the present manuscript is not fundamentally new to this respect.
- The article lacks quantitative analyses and discussions of the experiments. Its reading would be virtually unchanged were they not reported.
- The comparison between the simulations of the model and the experiments is too superficial to be convincing. Only snapshots of the systems are compared. The evolution of droplet size with time in the different regimes, the droplet velocity evolution with the energy input, for instance, are

accessible experimentally. How do these measurements compare to the simulations?

- Phase diagram. The discussion of the growth of droplids suggest that, at long time, coalescence and coarsening lead to immotile droplids. It is not clear whether the phase diagram of Fig. 2 characterize the steady state of the system or some transient. This point needs to be addressed and clarified early in the article.

- Some claims, in particular in the last section could/should be substantiated by deeper analyses: Page 13: how does the accumulation of non-absorbing particles at the interface increase with λ ?

Page 14: "this deviation originates from the varying velocities of active droplids at late times":

How does the velocity of droplids evolve in time? At short time, the exponent is already below $\frac{1}{2}$.

Page 14: "...temperatures quickly drop below T_c ": could this be shown?

Other comments:

- Page 2. "In contrast with equilibrium systems at thermodynamic equilibrium with their environment, active-matter systems can achieve complex forms of organization". Again this is an overstatement: complex structures do form at equilibrium (crystals, quasicrystals, emulsions etc.).

- Page 3, Eq. (3). Shouldn't the last term be a decay towards the temperature of the water bath?

- Figure 2. Is there a fundamental difference between phases II and III? Is there no excess of water around the molecules? What are the criteria used to distinguish between the two?

- Pages 11 and 12. The discussion about droplet growth is hard to follow. It goes not explicitly from discussing the case $\phi > 0.2$ to $\phi = 0.05$. The different cases could be displayed on the phase diagram of Fig. 2.

- Figure 4. The energy input λ could be given to improve the readability of the figures.

- Figure 4. The scale of ϕ is missing.

- Are the spacer particles affecting the phenomenology reported? How low is their density?

- Supplementary information Figure S3. What are the parameter values used for the hydrophobic particles?

Reviewer #3 (Remarks to the Author):

This paper presents a joint experimental and numerical study of a system that exhibits a two-way coupling between its constituents and their environment. The authors utilize a mixture of passive colloids and light-absorbing colloids that heat up upon illumination, in combination with a near-critical environment that locally phase separates upon a rise in temperature. They show that under certain circumstances, the colloids unify to form larger droplets that can acquire self-propulsion, thus resulting into "active droplids". They model the system theoretically by using a Cahn-Hilliard like description for the near-critical fluid, combined with the Brownian motion of the constituents coupled with the forces arising from concentration gradients and together with the dynamics of the temperature due to the illuminated source, diffusion and absorption. To map out the phase diagram for the system, they vary two parameters- 1) the relative concentration of the mixture and 2) the energy input given to the system. They parameterize the energy input in terms of the product of the density of the light absorbing colloids and the intensity of the illumination. Varying these two parameters in numerical simulations, they show that the theoretical model predicts all the states observed in the experiments. They further investigate the dynamics of the growth of the active droplids, measuring the droplid sizes vs time to uncover different scaling behaviors.

I find the paper to be novel, interesting, and a significant step forward for the active matter field. In particular, the achievement of two-way coupling between the particle behavior and their environment, allowing feedback between activity and the local environment, leads to emergent behaviors which are striking and novel. I agree with the authors that this could serve as a minimal model to provide some insight into biological systems, and opens doors for new design principles for applications of active materials. Thus, other than relatively minor comments below, I think the paper is worthy of publishing.

- In the 5th paragraph of the "Phase Diagram" section, the authors write "Because the dynamics of the particles is mainly controlled by the interaction of their surface with the local composition of the mixture, non-absorbing particles, which are less hydrophilic than their absorbing counterpart,

move towards the droplet's interface to reduce the total interfacial area of the system". This seems to suggest that the non-absorbing particles are moving to the interface because of their low hydrophilicity. However, parameters chosen in the simulation ($\alpha_a=0$, α_{na+ve}) explicitly add attraction to the interface only to the non-absorbing particles. It is unclear what aspects of the physics are emergent and what are explicitly put in the model. Could the authors elaborate on what the effect of the various parameters is in this regard?

- In the non-dimensional energy input parameter λ , the authors assume that the absorbed energy is linear in both active particle density and light intensity. This seems clear in a dilute solution, but it seems that multi-body effects could come into play within dense droplets. Have the authors ruled this out, or is this simply a reasonable first order approximation?

-While I agree with the authors that this could serve as a minimal model to provide insights into biological systems, I think the suggestion that it reveals principles of formation of membrane-less organelles requires at least some additional explanation.

- The composition order parameter used in the phase diagram, which seems to be $\langle \phi \rangle$ averaged over space, is also called ϕ in the text. It would be helpful if ϕ and $\langle \phi \rangle$ are distinguished.

- Figure 1 – While the authors provide all the information needed to back out the values of ϕ and λ to figure out where the snapshots lie on the phase diagram, it would be useful to have the value of these parameters in the figure caption.

- The second paragraph of "Model and Simulations" defines phase A to be water and phase B to be 2,6-lutidine and assigns $\phi = \pm 1$ to the phases A and B respectively. However, all the other parts of the manuscript (Figure 1, the sign of β_s being negative) seem to indicate that $\phi = -1$ (phase B) corresponds to water. Can the authors explain or revise the definition in this paragraph?

Similarly, the article reads "We describe the impact of the hydrophilicity of the light-absorbing and non-absorbing particles on the dynamics of the fluid with an additional term including a δ -function at the particle positions, whose strength is given by A_a and A_{na} , respectively." It would be useful to further explain how these terms affect the ϕ dynamics.

- In Equation 3, the coupling of the sample to the external water heat bath should be $-k_d (T-T_0)$?

-Figure 3a – The legend label "passive droplets" is confusing. Perhaps it would better to label it as immotile droplets?

-Figure 4 – It would be useful to add colorbars for the simulation plots.

typos:

- "Experimental observations of active droplets" section – 3rd paragraph – 1st sentence – there is a missing or extra parenthesis.

-Additional Information – "Details on the simulation model" section – First paragraph – "The evolution of the conserved order parameter (composition of the two components) is [than then] given by the Cahn-Hilliard equation".

Reply to the comments of the reviewers on the manuscript “Active Droploids”

Reviewer #1 (Remarks to the Author):

In the manuscript “Active droploids”, the authors report on the realization of a two-way coupling between active particles and their environment, where active particles act back on the environment giving rise to the formation of superstructures that they call “droploids”. Droploids are realized by light-activated colloids placed in a near-critical environment. When illuminated by laser light, these colloids absorb the light, locally heat up the surrounding environment, and induce a local phase separation leading to the emergence of droplets. The self-propulsion mechanism of symmetry-broken assemblies has been demonstrated earlier by the authors (J. Chem. Phys., 2019, 150, 094905) and others (Chem. Comm. 2018, 54, 11933). The authors employ these active assemblies to act as an engine in the droplet-colloids superstructures like the recently demonstrated active droplets (Soft Matter, 2020, 16, 1359 and Nat. Phys., 2021, 17, 260).

The authors experimentally realize droploids and theoretically model the observed phenomenology. The diagram of the dynamic states has been successfully captured by the simulations, as well as the aggregation dynamics of the droplets. The paper is clearly written, and conclusions seem to be well supported by the presented data.

We thank the reviewer very much for taking the time to carefully read the manuscript, for the positive assessment and for providing useful suggestions for improvement. In the following we reply point by point to the reviewer’s remarks.

At the end of the paper (conclusions section), the authors got a bit bombastic with the claim of “a novel class of active matter” associated with the droploids. Active droplets - combining droplets and active particles - have been around both in simulations and experiments for quite some time.

Following the reviewer’s comments we have revised the corresponding text passage by emphasizing the new findings in the present work compared to existing works combining droplets and active matter. The corresponding paragraph now reads:

“Our results show that a *two-way coupling* between the motion of colloidal particles and the *dynamics of their environment* creates a route towards a novel class of active superstructures. These structures hinge on mutually-coupled structure formation processes of the colloids, which form an engine, and the surrounding solvent, which phase separates in regions of high colloidal density and encapsulates the engine within a droplet shell. Our results create a bridge between the physics of active colloids and droplets and provide fundamental insights into the role of feedback for the emergence of ordered active superstructures, which opens up new possibilities for active-matter research to investigate two-way feedback loops in other systems and to create light-activated biomimetic materials.”

A few points that need further clarification:

In relation to the phoretic motion of non-absorbing particles discussed on Page 10 and Fig. 2c and 2d. Both absorbing and non-absorbing particles are hydrophilic. It is not clear why hydrophilic non-absorbing particles should go to the interface at elevated temperatures. Is it the difference in the degree of hydrophilicity between those two species of particles that promotes their motion towards the interface?

As the reviewer writes, it is the degree of hydrophilicity that determines where the particle is placed. The system places the particles where they maximally reduce its free energy. Our absorbing particles are strongly hydrophilic and therefore prefer and stay in the water-rich phase. The non-absorbing particles, on the other hand, are only very weakly hydrophilic and behave similarly to completely amphiphilic particles, which have no preference for any of the two phases. Accordingly, the system can reduce its free energy by placing these particles at the water-lutidine interface such that they reduce the water-lutidine surface area and hence the total free energy of the water-lutidine, particle-water and particle-lutidine interfaces together.

To confirm the applicability of this picture to our active droplet experiments, we have done additional experiments using Silica particles whose surface has been modified using (3-mercaptopropyl)trimethoxysilane (MPTS) creating a highly hydrophilic surface. In those cases the particles remained inside the water-rich droplets without going to the interface (Fig. 1.1 a). In contrast, by modifying the surface based on tertbutyldiphenylsilane (TBDS), a weak hydrophobicity is reached such that particles move towards the water-lutidine interface surrounding a droplet and stay there (Fig. 1.1 b). Analogously, in our active-droplet experiments weakly hydrophilic particles approach the interface from within the water-rich droplet. We thank the reviewer for inspiring these additional explorations.

Figure 1.1: **a** Strongly hydrophilic Silica particles ($a = 1.25\mu\text{m}$, white) and light-absorbing particles ($a = 0.5\mu\text{m}$, dark) stay immersed inside water-rich droplets. **b** Weakly hydrophobic silica particles ($a = 0.6\mu\text{m}$, white) move towards the droplet interface while the hydrophilic absorbing particles remain immersed.

Also, what happens with droplets when you let the system relax below critical temperature?

If the temperature drops below T_c , phase separation does not occur. The mixture then returns to a phase in which water and 2,6-lutidine mix homogeneously and existing droplets dissolve. This can be observed in the following snapshots from simulations (Fig. 1.2 a,b) and from experiments (Fig. 1.2 c,d). When the light source is on, the temperature near the absorbing particles is above T_c , allowing them to form droplets. When the light source is off, the temperature quickly drops to the value of the water heat bath T_0 and the droplets dissolve.

Consequently, when light illumination is switched off, we no longer observe any directed motion of the structures. Thus, the mean velocity of all particles drops abruptly as soon as particles are no longer illuminated, indicating how the molecules lose their self-propulsion ability.

When switching the light source on again, the active droplets emerge again such that their formation process can be controlled by light and is fully reversible.

Figure 1.2: **a** Simulation snapshots of the composition order parameter ϕ and the temperature field T when the light source is on (left) and off (right) and **b** time-evolution of the mean particle velocity when the light illumination is switched suddenly off at $t = 10\text{s}$. **c** Experimental

snapshots of droplids when the light source is switched on and $d \approx 2.1$ s after it has been switched off where droplets have completely dissolved.

Reviewer #2 (Remarks to the Author):

The authors report on the light-triggered assembly of colloidal clusters in near-critical environments. The manuscript focuses on describing the phenomenology of the system when the composition of the environment (its distance from the critical mixture) and the heat input are varied.

I have several reservations about the manuscript which in the present form lack the completeness expected for publication in Nature Communications. While the system displays rich behaviours, its characterization would deserve to be more thorough. Now, it appears somewhat like an addendum to a previous publication by the authors (Ref 31. <https://aip.scitation.org/doi/10.1063/1.5079861>).

The very claim that no environment feedback was ever at play in active matter system (bottom of page 2) is inaccurate. What about the role of boundaries in active system (<https://www.nature.com/articles/nphys3876> for a recent example)? What about environment-mediated interactions between active particles (<https://doi.org/10.1103/PhysRevLett.123.208101> for a recent example)? The situation in the present manuscript is not fundamentally new to this respect.

We thank the reviewer for his/her careful reading and for providing numerous useful suggestions to further improve the manuscript. In the following we reply point by point to his/her remarks.

We agree with the reviewer that the concept feedback in active systems has been studied previously. However, the crucial novelty in the present work is a full *two-way* feedback between a system and an environment which both dynamically respond to changes of the other component, i.e. they both *feature an intrinsic time-evolution leading to structure formation in both components*. Such a feedback loop between colloids and the surrounding solvent is – to our knowledge – fundamentally new. The key achievement of the present work is to show that this new ingredient leads to a previously unknown type of structure: active droplids — featuring a light-powered self-assembled colloidal engine. We thank the reviewer for bringing up this point and have modified our manuscript to specify this point.

The reviewer mentions two very interesting references that we would like to further discuss here in more detail.

The first reference (<https://www.nature.com/articles/nphys3876>) investigates the important influence of boundaries on the nematic arrangement of their system. However, this example displays a one-way coupling only, i.e. from the boundary to the active nematics system, whereas the boundaries do not respond to the dynamics of the nematic.

In the second reference (<https://doi.org/10.1103/PhysRevLett.123.208101>), where two filaments mutually influence each other via hydrodynamic interactions, the environment essentially allows the filaments to synchronize their dynamics. However, the environment here responds instantaneously to the motion of the filaments and does not show any intrinsic dynamics and, in particular, no structure formation. A similar type of environmentally mediated coupling occurs in agents which are e.g. visually, acoustically or chemically coupled.

Thus, both examples do not display the novel two-way coupling between the system and its environment as we do observe in our experiments and simulations. We have clarified this now further in our current manuscript by including the following two paragraphs to the introduction of the revised manuscript:

“In all these examples, and other typical active matter systems, the environment serves as a continuous free-energy source which can also mediate effective interactions between active agents, such as hydrodynamic interactions synchronizing filaments [26] or interactions based on visual perception [27], acoustic signals [28] or chemical fields [29] - but it does typically not show an intrinsic dynamics which adapts to the dynamics of the active agents. In contrast, biological systems often show a two-way coupling, which is involved e.g. in homeostasis, gene-expression regulation, and structure formation [30,31], calling for synthetic realizations to enable a controllable exploration of two-way feedback coupled systems.”

“Note that, crucially, both the colloids' and droplets' system exhibit an intrinsic self-dynamics that is fundamental to the droplets' structure formation and self-propulsion ability.”

- The article lacks quantitative analyses and discussions of the experiments. Its reading would be virtually unchanged were they not reported.

Following the reviewer's comments we have extensively reanalyzed our experimental data in order to quantitatively compare the experiments with the simulations. The results of this comparison are presented in the strongly revised Fig. 2 and in particular Fig. 3, which we will discuss in detail in the following replies to the reviewer's more specific points.

- The comparison between the simulations of the model and the experiments is too superficial to be convincing. Only snapshots of the systems are compared. The evolution of droplet size with time in the different regimes, the droplet velocity evolution with the energy input, for instance, are accessible experimentally. How do these measurements compare to the simulations?

Following the reviewer's comments and questions we have added specific experimental data points to the phase diagram (Fig. 2a of the revised manuscript, reprinted below) for different relative concentrations Φ_0 . The comparison shows that the experiments and the simulations are consistent and in good quantitative agreement with each other.

Fig. 2a of the revised manuscript: Phase diagram as a function of the net energy input λ and the averaged relative concentration difference from the critical point Φ_0 . The evaluated state points from the experiment (red) and the simulations (black) are indicated by crosses (purple – disordered phase), triangles (yellow – active molecules), filled circles (green – active droplets), and empty circles (blue – droplets with particles at the interface).

To further quantify the relation between our model and experiments, in Fig. 3 of the revised manuscript (reprinted below) we show the results of a newly developed analysis method of our experiments to quantify the droplet dynamics in comparison with our simulations, as suggested by the reviewer. Since the relevant data which is required for a detailed quantitative comparison with our simulations was not fully accessible based on standard tracking methods, we have re-analyzed our simulations in much detail based on in-house developed machine learning algorithms (using the DeepTrack 2.0 software), as we explain in the supporting information of the revised manuscript. This method allows us to identify and track the motion of droplets over time providing us with detailed information of their location and speed as well as of their size and shape. In particular, this has allowed us to directly compare the ensemble-averaged time-evolution of the droplet size in our experiments with our simulations as suggested by the reviewer. In Fig. 3a of the revised manuscript (reprinted below) one can now see that the exponent of time in simulations ($t^{0.42}$ for active droplets, $t^{0.31}$ for passive droplets) corresponds well with our experimental data ($t^{0.41}$ for active droplets, $t^{0.35}$ for passive droplets) despite their relatively large error bars (indicated by the coloured background with the mean plotted as dotted line). The size of the error bars is not surprising as the colloids used in our experiments are not perfectly identical, leading to significant fluctuations of the droplet size from experiment to experiment.

To further compare our experiments with the developed model, we now also provide new information on the travelled distance of active droplets over time as well as their velocity distribution (averaged over all considered light intensities for statistical reasons). Also here, the experimental results are in good agreement with our simulations. Notice that the long-time evolution of the ensemble-averaged speed of the active droplets over time is comparatively difficult to compare as the droplets are frequently moving out of frame and are merging with other droplets.

Finally, we have also added experimental data points to Fig. 3e of the revised manuscript (reprinted below) to test the predictions for the fraction of non-absorbing colloids which are attaching to the droplet interface, depending on the energy input λ . These data points confirm the predicted trend that non-absorbing particles increasingly accumulate at the droplet interface as λ increases.

Fig. 3 of the revised manuscript: Droplet velocity and growth over time. **a** Average droplet size of active droplets (green) and immotile droplets (blue) over time calculated from experiments (dotted) and simulations (solid) for $\phi_0 = 0.05$. The inset shows the delay in the formation of an immotile droplet at early times for an off-critical composition of $\phi_0 = 0.25$. **b** Total travelled distance of an active droplet over time calculated from experiments (dotted) and simulations (solid). **c** Simulated droplet velocity over time. **d** Velocity distribution of active droplets in experiments (left) and simulations (right). **e** Simulated mean velocity of active droplets after 30 s of light illumination (black curve) and fraction of non-absorbing particles located at the interface of the droplets N_{na}^{int}/N_{na} (grey curve simulations, grey dots experimental data) as a function of the energy input λ for $\phi_0 = 0.05$. The full list of simulation parameters is provided in the methods.

- Phase diagram. The discussion of the growth of droplets suggest that, at long time, coalescence and coarsening lead to immotile droplets. It is not clear whether the phase diagram of Fig. 2 characterize the steady state of the system or some transient. This point needs to be addressed and clarified early in the article.

We thank the reviewer for this comment and agree that this is an important point which we now explicitly address in the manuscript. In particular, following the reviewer’s comment we have studied the time-evolution of the “phase boundaries” for various energy inputs λ at fixed composition $\phi_0 = 0.1$ over time. The result is shown in Fig. 2.1 below. It can be seen that the phase boundaries change initially but then converge to a plateau and do not change further, at least not on experimentally relevant time scales.

In the SI we have added the figure below and also specify the exact criteria which we use to define the different phases and colors (see also the discussion further below).

Figure 2.1: Phase diagram as a function of the net energy input λ and the time for a fixed averaged relative concentration difference from the critical point $\phi_0 = 0.1$.

- Some claims, in particular in the last section could/should be substantiated by deeper analyses:

Page 13: how does the accumulation of non-absorbing particles at the interface increase with λ ?

In our previous manuscript we have shown in simulations that the number of non-absorbing particles at the interface N_{na}^{int} increases with increasing the illumination strength k_0 . As suggested by the reviewer we now show the same quantity as a function of λ , which allows for a more direct link to the phase diagram.

In addition, we have added experimental data to the relevant plot to confirm the observed trend for N_{na}^{int} increasing with λ . (Note, that the relatively large fluctuations of the experimental data points are caused by the different densities of absorbing particles which have been used in the underlying experiments.)

Figure 2.2: Fraction of non-absorbing particles located at the interface of the droplets N_{Na}^{int} / N_{Na} as a function of the energy input λ for $\phi_0 = 0.05$.

Page 14: "this deviation originates from the varying velocities of active droplids at late times": How does the velocity of droplids evolve in time? At short time, the exponent is already below $\frac{1}{2}$.

To show how the velocity of the droplids evolves in time, we have calculated it from simulation data, whose results are plotted in Fig. 2.3 below. It can be seen that droplids have a finite

speed of about $1.4\mu\text{m/s}$ immediately after their formation and further speed up as time evolves until they reach a speed maximum of about $1.9\mu\text{m/s}$ after about 30 seconds. This acceleration is caused by the absorption of additional colloids entering the droplid which become part of the colloidal engine driving the droplid forward.

At longer times, however, the droplids slow down again but still move ballistically (while their sizes grow with an exponent smaller than $\frac{1}{2}$). Ultimately, they would reach a state where light-absorbing and non-absorbing particles form a major cluster of randomly arranged absorbing and non-absorbing particles within the droplet shell, which all contribute to the friction the droplid experiences but do not significantly support self-propulsion. The growth of the colloidal clusters within the droplet shells also enhances the temperature locally which reduces the degree of demixing and supports the slow-down of the active droplids. We have added a corresponding remark also to the manuscript.

Figure 2.3: Simulated mean velocity of active droplids over time.

Page 14: "...temperatures quickly drop below T_c ": could this be shown?

Phase separation does not occur when the temperature falls below T_c . Existing droplets then disintegrate as the mixture returns to a phase in which water and 2,6-lutidine are homogeneously mixed and the mean particle velocity rapidly decreases. The following Fig. 2.4 demonstrates this both for simulations (panels a,b) and experiments (panel c,d).

In the reverse case, when the light-source is suddenly switched on and the temperature increases to a value above T_c droplets quickly (re)form and the mean-particle velocity rapidly increases before decreasing to a certain characteristic value (Fig. 2.5).

Overall, the formation of active droplids is fully light-controlled. When the light source is repeatedly switched on and off, the droplids alternatingly disintegrate and reform.

Figure 2.4: **a** Simulation snapshots of the composition order parameter ϕ and the temperature field T when the light source is on (left) and off (right) and **b** time-evolution of the mean particle velocity when the light illumination is switched suddenly off. **c** Experimental snapshots of droplets when the light is switched on and **d** 2.1 seconds after it has been switched off where droplets have completely dissolved.

The calculated mean particle velocity is depicted in Fig. 2.5. At $t = 0$ the motion of the particles is governed solely by diffusion. As soon as the light illumination is switched on and consequently the structures are formed, one can observe how the mean velocity per particle increases abruptly. The increased value immediately after switching on is due to the sudden accumulation of particles and formation of molecules. From then on, the formed structures move with increased velocity.

Figure 2.5: Mean particle velocity when turning on light illumination after five seconds.

Other comments:

- Page 2. *“In contrast with equilibrium systems at thermodynamic equilibrium with their environment, active-matter systems can achieve complex forms of organization”. Again this is an overstatement: complex structures do form at equilibrium (crystals, quasicrystals, emulsions etc.).*

We agree with the reviewer and have clarified our statement in the manuscript. Equilibrium structures are determined by the optimal compromise between entropy maximization and energy minimization which in particular rules out any structures showing persistent dynamics. In contrast, the droplets which we observe are active and indeed show persistent dynamics. In addition, they also have a non-trivial length scale which is not identical with the length scale of their interparticle interactions. (In equilibrium however, the length scale of the structures which occur typically have a length scale which is identical to the length scale of the interactions of its components.)

“In contrast to equilibrium systems, active-matter systems evade the rules of equilibrium thermodynamics by continuously dissipating energy obtained from their environment and converting part of it into a persistent motion at the level of individual constituents.”

- Page 3, Eq. (3). *Shouldn't the last term be a decay towards the temperature of the water bath?*

Yes, that is absolutely correct. The last term is a decay towards the temperature of the water heat bath T_0 . We have corrected this typo in the manuscript and thank the reviewer for pointing it out.

- Figure 2. Is there a fundamental difference between phases II and III? Is there no excess of water around the molecules? What are the criteria used to distinguish between the two?

We thank the reviewer for this interesting question. The difference between phases II and III is that only in phase III the colloids are surrounded by a water-rich droplet with a size beyond that of typical wetting layers and with a clearly identifiable interface.

More quantitatively, one can discriminate between phases II and III by defining the following criterion

$$\frac{A_{\text{Droplid}} - 4\pi r^2}{4\pi r^2} > 1,$$

which requires that the area of a water-rich droplid is significantly larger than the surface area of the colloids within a droplid (here we choose the surface area of eight colloids which is about twice the typical number of colloids within an early-time droplid in our simulations and typical for droplids at late stages.).

We checked this criterion for the different parameter combinations in the phase diagram in Fig. 2 of the manuscript and plotted it in the following figure. The left y-axis (black) shows the left side of the above criterion, which is satisfied for the structures in the green and blue regions where we can find active and passive droplids.

To additionally distinguish between "active" and "passive" droplids, the average velocity of all particles (including active structures but also single, possibly passive, particles) v_p is given on the right y-axis (gray). If this average velocity clearly exceeds that of passive Brownian particles, i.e. if $v_p > 0.8 \mu\text{m}/\text{s}$ (for the used sampling rate), the structures are in the region of active molecules or active droplets.

Figure 2.6: Area covered by water that exceeds the occupied area of immersed particles and mean particle velocity for different values of ϕ_0 and λ .

Based on the criteria described, it can be seen that there is actually a crossover between the different (nonequilibrium) phases instead of an abrupt sharp transition.

We have updated the phase diagram in Fig. 2a of the revised manuscript and specifically define the meaning of the used colors (based on Fig. 2.6 which has been added to the supplementary information). Instead of sharp color changes we now also indicate the crossover regimes by color gradients.

- Pages 11 and 12. The discussion about droplet growth is hard to follow. It goes not explicitly from discussing the case $\phi_0 > 0.2$ to $\phi_0 = 0.05$. The different cases could be displayed on the phase diagram of Fig. 2.

We thank the reviewer very much for this comment. As suggested by the reviewer, we have marked the different cases in the phase diagram of Fig. 2a of the manuscript. We have also restructured and extended the section on droplet dynamics and growth:

“Using a composition of $\phi_0 = 0.25$ (marked with a red-bordered 1 in Fig. 2) which is far away from the critical composition, we observe that droplets form only after a significant initial delay.”

“For concentrations around $\phi_0 = 0.05$ (marked with a red-bordered 2 in Fig. 2a), after the droplets have formed, we observe a transition from nucleation and growth to coarsening.”

“We observe that the size of active droplets grows as $L \sim t^{0.42}$ (marked with a red-bordered 3 in Fig. 2a and shown in Fig. 3a) which is significantly faster than the observed $L \sim t^{0.31}$ growth law for passive droplets. The accelerated growth is a direct consequence of the ballistic motion of the active droplets which allows them to recruit additional colloids faster than the passive droplets.”

- Figure 4. The energy input λ could be given to improve the readability of the figures.

We have added the values of the energy input λ for the different scenarios in Fig. 4.

- Figure 4. The scale of ϕ is missing.

We thank the reviewer for pointing this out. We have added colorbars for ϕ in Fig. 4.

- Are the spacer particles affecting the phenomenology reported? How low is their density?

The density of spacer particles is sufficiently small ($c \ll 5\%$) to not significantly influence the phenomenology observed in the experiments, which we have now also added as information under methods in the revised manuscript:

“We use spacer particles (silica microspheres, microParticles GmbH) with a radius $R = 0.85 \pm 0.02 \mu\text{m}$ for constant separation but with a concentration $c \ll 5\%$, in order to not interfere with the observed phenomena.”

- Supplementary information Figure S3. What are the parameter values used for the hydrophobic particles?

The hydrophobic particles which we use are similar to the hydrophilic non-absorbing particles: they have a radius of $R=0.49$ and similar density of $\rho=2\text{g/cm}^3$. The experiments have been performed at the same conditions with $\phi_0=0.05$ and $T_0=32.5^\circ\text{C}$. We have added this information to the supplementary information.

Reviewer #3 (Remarks to the Author):

This paper presents a joint experimental and numerical study of a system that exhibits a two-way coupling between its constituents and their environment. The authors utilize a mixture of passive colloids and light-absorbing colloids that heat up upon illumination, in combination with a near-critical environment that locally phase separates upon a rise in temperature. They show that under certain circumstances, the colloids unify to form larger droplets that can acquire self-propulsion, thus resulting into “active droplids”. They model the system theoretically by using a Cahn-Hilliard like description for the near-critical fluid, combined with the Brownian motion of the constituents coupled with the forces arising from concentration gradients and together with the dynamics of the temperature due to the illuminated source, diffusion and absorption. To map out the phase diagram for the system, they vary two parameters- 1) the relative concentration of the mixture and 2) the energy input given to the system. They parameterize the energy input in terms of the product of the density of the light absorbing colloids and the intensity of the illumination. Varying these two parameters in numerical simulations, they show that the theoretical model predicts all the states observed in the experiments. They further investigate the dynamics of the growth of the active droplids, measuring the droplid sizes vs time to uncover different scaling behaviors.

I find the paper to be novel, interesting, and a significant step forward for the active matter field. In particular, the achievement of two-way coupling between the particle behavior and their environment, allowing feedback between activity and the local environment, leads to emergent behaviors which are striking and novel. I agree with the authors that this could serve as a minimal model to provide some insight into biological systems, and opens doors for new design principles for applications of active materials. Thus, other than relatively minor comments below, I think the paper is worthy of publishing.

We thank the reviewer for his/her very careful reading of the manuscript, for the positive assessment and for providing several suggestions to further improve the paper. In the following we reply point by point to his/her remarks.

-In the 5th paragraph of the “Phase Diagram” section, the authors write “Because the dynamics of the particles is mainly controlled by the interaction of their surface with the local composition of the mixture, non-absorbing particles, which are less hydrophilic than their absorbing counterpart, move towards the droplet’s interface to reduce the total interfacial area of the system”. This seems to suggest that the non-absorbing particles are moving to the interface because of their low hydrophilicity. However, parameters chosen in the simulation ($\alpha_a=0$, α_{na+ve}) explicitly add attraction to the interface only to the non-absorbing particles. It is unclear what aspects of the physics are emergent and what are explicitly put in the model. Could the authors elaborate on what the effect of the various parameters is in this regard?

We thank the reviewer for bringing up this point. As the reviewer writes, the absorbing particles are strongly hydrophilic and prefer the water-rich droplets, whereas non-absorbing particles are only weakly hydrophilic. Therefore the system can reduce its free energy by placing the non-absorbing particles at the interface between water-rich droplets and the lutidine-rich phase.

This can easily be quantified based on the change of the surface free energy, ΔG , when displacing a particle from the water-rich phase symmetrically to the droplet-interface.

$\Delta G = 2\pi R^2\gamma_{PW} + 2\pi R^2\gamma_{PL} - \pi R^2\gamma_{WL} - 4\pi R^2\gamma_{PW} = -2\pi R^2\gamma_{PW} + 2\pi R^2\gamma_{PL} - \pi R^2\gamma_{WL}$, where R is the particle radius, γ_{WL} is the interfacial tension at the water-lutidine interface, γ_{PL} is the interfacial tension at the particle-lutidine and γ_{PW} at the particle-water interface.

For the strongly hydrophilic light-absorbing particles we have $\gamma_{PL} \gg \gamma_{PW}$ and $\gamma_{PL} > \gamma_{WL}$ and therefore $\Delta G > 0$, which keeps them in the water-rich droplets. In contrast, the non-absorbing particles are only weakly hydrophilic we have $\gamma_{WL} \gg \gamma_{PL} - \gamma_{PW}$ (this is similar to completely amphiphilic particles where $\gamma_{PL} = \gamma_{PW}$) and thus $\Delta G < 0$, causing the non-absorbing particle to move to the droplet’s surface.

In our theoretical model, we are mainly interested in the motion of the colloids and have therefore included the net effect of the attraction which results from these considerations based on the term of the form $\alpha_s \nabla(\nabla\phi)^2$.

Following the reviewer’s question, we now explain the reasoning underlying this term in the description of the model:

“The second term, which is proportional to $\nabla(\nabla\phi)^2$, induces motion towards interfaces (where $\nabla\phi^2$ is large) essentially for the non-absorbing particles. This term describes the tendency of the weakly hydrophilic particles to move towards the water-lutidine interface in order to reduce the interfacial area of the water-lutidine interface and hence the total interfacial free energy of the system.”

- In the non-dimensional energy input parameter λ , the authors assume that the absorbed energy is linear in both active particle density and light intensity. This seems clear in a dilute solution, but it seems that multi-body effects could come into play within dense droplets. Have the authors ruled this out, or is this simply a reasonable first order approximation?

We fully agree with the reviewer. To emphasize that this is an approximation strictly valid in a relatively diluted regime, we have now changed the relevant sentence in the manuscript:

“Overall, this permits us to introduce a concise measure of the energy input under the approximation that the adsorbed energy scales linearly with the density of the absorbing particles and the irradiated light intensity as $\lambda = \frac{\rho_a k_0}{T_0 k_d}$.”

-While I agree with the authors that this could serve as a minimal model to provide insights into biological systems, I think the suggestion that it reveals principles of formation of membrane-less organelles requires at least some additional explanation.

We agree with the reviewer and have now added more clarification to why our results could reveal more insights into the formation of membrane-less organelles:

“Since it is based on the two-way interaction between colloidal particles and a near-critical environment, the involved mechanism of colloidal self-encapsulation and activation might also serve as a useful framework to recreate and explore aspects of fundamental biological processes in a well-controllable synthetic colloidal minimal systems. Examples might comprise e.g. processes which are involved in the compartmentalization inside the cytoplasm and the formation of membrane-free organelles, which share various features with liquid droplets and do not need a stabilizing lipid bilayer to maintain themselves.”

- The composition order parameter used in the phase diagram, which seems to be $\langle \phi \rangle$ averaged over space, is also called ϕ in the text. It would be helpful if ϕ and $\langle \phi \rangle$ are distinguished.

We thank the reviewer for raising this issue. We have renamed the averaged composition in the phase diagram and similarly corrected the manuscript in all corresponding further places.

- Figure 1 – While the authors provide all the information needed to back out the values of ϕ and λ to figure out where the snapshots lie on the phase diagram, it would be useful to have the value of these parameters in the figure caption.

We added the values of ϕ and λ to the caption of Fig.1.

- The second paragraph of “Model and Simulations” defines phase A to be water and phase B to be 2,6-lutidine and assigns $\phi = \pm 1$ to the phases A and B respectively. However, all the other parts of the manuscript (Figure 1, the sign of β_s being negative) seem to indicate that $\phi = -1$ (phase B) corresponds to water. Can the authors explain or revise the definition in this paragraph?

We thank the reviewer for pointing out this problem. Of course, $\phi < 0$ corresponds to water-rich regions and $\phi > 0$ to lutidine-rich regions. Accordingly, we have corrected this paragraph in the manuscript.

Similarly, the article reads “We describe the impact of the hydrophilicity of the light-absorbing and non-absorbing particles on the dynamics of the fluid with an additional term including a δ -function at the particle positions, whose strength is given by A_a and A_{na} , respectively.” It would be useful to further explain how these terms affect the ϕ dynamics.

We thank the reviewer for this comment and have now explained the influence of the terms which are led by the coefficients A_a and A_{na} in more detail. The relevant text passage now reads:

“To describe the net effect of the accumulation of water at the hydrophilic surfaces of the colloids, we include a (point-like) source term for the solvent-composition at the position of each particle. The coefficients A_a and A_{na} are chosen such that they account for the strong and weak hydrophilicity of the absorbing and non-absorbing particles respectively. As a result, the water concentration slightly increases at the location of each particle. Note that this increase alone does of course not initiate phase separation, but it biases the location where water-rich droplets occur once phase separation takes place. “

- In Equation 3, the coupling of the sample to the external water heat bath should be $k_d(T-T_0)$?

Yes, that is absolutely correct. The last term is a decay towards the temperature of the water heat bath T_0 . We have corrected this typo in the manuscript.

-Figure 3a – The legend label “passive droplids” is confusing. Perhaps it would better to label it as immotile droplets?

We agree with the reviewer and have changed the label in Fig. 3.

-Figure 4 – It would be useful to add colorbars for the simulation plots.

Thanks for this suggestion. We have now added colorbars to the simulation plots in Fig. 4.

typos:

-“Experimental observations of active droplets” section – 3rd paragraph – 1st sentence – there is a missing or extra parenthesis.

-Additional Information – “Details on the simulation model” section – First paragraph – “The evolution of the conserved order parameter (composition of the two components) is [than then] given by the Cahn-Hilliard equation”.

We thank the reviewer for noting these typos and have corrected them accordingly.

REVIEWERS' COMMENTS

Reviewer #1 (Remarks to the Author):

The authors constructively addressed all the comments from the reviewers. The manuscript is considerably improved now and could be recommended for acceptance.

Reviewer #2 (Remarks to the Author):

I thank the authors for their thorough consideration of my comments. The revised manuscript is more accurate and improved by the analysis of experiments and their comparison with computational results. In its present form, it now meets the standards for publication in Nature Communications.

I only have minor comments:

- About the "two-way coupling". As pointed out by the authors in the reply, the active-droplets specificity lies in the formation of "structures" (droplets) in the environment, rather than on a very generic two-way coupling. The authors may want to make this point clear in the introduction. On page 2, it is not clear to me what "intrinsic self-dynamics" means.
- I really appreciate that experiment results are compared as much as possible with numerical ones in the revised manuscript. However, the fitting coefficient C used in Figure 2a and Figure 3e differ by a factor 3. What is the reason for this? Mentioning this difference explicitly would strengthen the Article.
- Figure 3. What the error bars represent should be added within the caption.

Reviewer #3 (Remarks to the Author):

It appears to me that the authors have adequately responded to the reviewer comments. In particular, they now better emphasize the two-way feedback between their system and the external environment, which to my knowledge is new in the field of active matter. They have also included extensive new results showing comparison between their simulations and experiments, which significantly increases support for the idea that the simulations capture the key physical ingredients. Thus in my opinion the article is ready for publication.

Reply to the referee reports on manuscript NCOMMS-21-02947A

Reply to referee 1:

The authors constructively addressed all the comments from the reviewers. The manuscript is considerably improved now and could be recommended for acceptance.

We would like to thank the referee very much for re-reading the manuscript and for recommending its publication in Nature Communications.

Reply to referee 2:

I thank the authors for their thorough consideration of my comments. The revised manuscript is more accurate and improved by the analysis of experiments and their comparison with computational results. In its present form, it now meets the standards for publication in Nature Communications.

I only have minor comments:

We would like to thank the referee very much for re-reading the manuscript, for concluding that the manuscript now meets the standard of Nature Communications and for the remaining minor comments. We have followed all corresponding suggestions.

- About the "two-way coupling". As pointed out by the authors in the reply, the active-droplets specificity lies in the formation of "structures" (droplets) in the environment, rather than on a very generic two-way coupling. The authors may want to make this point clear in the introduction.

On page 2, it is not clear to me what "intrinsic self-dynamics" means.

We thank the referee for these useful comments. Following the reviewer's suggestion, we now explicitly mention the structure formation taking place in the environment in the following two revised sentences. The relevant sentences now read:

"Here, we experimentally realize and theoretically model a versatile feedback loop between light-activated colloids and their near-critical environment. This leads to structure formation both on the level of the colloids and their environment and results in the formation of a new type of self-propelling superstructures."

"To realize this feedback loop and the corresponding structure formation processes in the environment, we suspend a mixture of colloids in a near-critical solvent."

Following the referee's comment, we have also rewritten the sentence which contained the term "intrinsic self-dynamics". It now reads:

Note that, crucially, both the colloids and the droplets continuously evolve in time, rather than adiabatically following the respective other component, which is fundamental to the droplets' structure formation and self-propulsion ability.

- I really appreciate that experiment results are compared as much as possible with numerical ones in the revised manuscript. However, the fitting coefficient C used in Figure 2a and Figure 3e differ by a factor 3. What is the reason for this? Mentioning this difference explicitly would strengthen the Article.

We thank the referee for pointing out this detail. We now address it explicitly in the manuscript as follows:

“Note that the fitting factor C is different here than in Fig. 2 because the present measurements are based on a fixed concentration of $\phi_0=0.05$, whereas those in Fig. 2 have been taken at various concentrations up to $\phi_0=0.2$.”

- Figure 3. What the error bars represent should be added within the caption.

We thank the reviewer for raising this issue and have added this accordingly in the caption:

“The shaded area represents the standard deviation.”

Reply to referee 3:

It appears to me that the authors have adequately responded to the reviewer comments. In particular, they now better emphasize the two-way feedback between their system and the external environment, which to my knowledge is new in the field of active matter. They have also included extensive new results showing comparison between their simulations and experiments, which significantly increases support for the idea that the simulations capture the key physical ingredients. Thus in my opinion the article is ready for publication.

We would like to thank the referee very much for reading the manuscript and the previous referee reports and for recommending its publication in Nature Communications.